# Oxime-Linked Peptide–Daunomycin Conjugates as Good Tools for Selection of Suitable Homing Devices in Targeted Tumor Therapy: An Overview

**DOI:** 10.3390/ijms25031864

**Published:** 2024-02-03

**Authors:** Gábor Mező, Jacopo Gomena, Ivan Ranđelović, Endre Levente Dókus, Krisztina Kiss, Lilla Pethő, Sabine Schuster, Balázs Vári, Diána Vári-Mező, Eszter Lajkó, Lívia Polgár, László Kőhidai, József Tóvári, Ildikó Szabó

**Affiliations:** 1HUN-REN-ELTE Research Group of Peptide Chemistry, 1117 Budapest, Hungary; jacopo.gomena@gmail.com (J.G.); dokuslevente@gmail.com (E.L.D.); lilla.petho@chem.elte.hu (L.P.); sabine.schuster83@gmail.com (S.S.); diusmezo@gmail.com (D.V.-M.); ildiko.szabo.elte.ttk@gmail.com (I.S.); 2Institute of Chemistry, ELTE, Eötvös Loránd University, 1117 Budapest, Hungary; 3Department of Experimental Pharmacology and the National Tumor Biology Laboratory, National Institute of Oncology, 1122 Budapest, Hungary; randelovic.ivan@oncol.hu (I.R.); vari.balazs@oncol.hu (B.V.); tovari.jozsef@oncol.hu (J.T.); 4Department of Organic Chemistry and Technology, Budapest University of Technology and Economics, 1111 Budapest, Hungary; 5School of Ph.D. Studies, Doctoral School of Pathological Sciences, Semmelweis University, 1085 Budapest, Hungary; 6Department of Genetics, Cell- and Immunobiology, Semmelweis University, 1089 Budapest, Hungary; lajesz@gmail.com (E.L.); gorlicze@gmail.com (L.P.); kohlasz2@gmail.com (L.K.)

**Keywords:** targeted tumor therapy, peptide–drug conjugates, homing peptides, daunomycin, oxime linkage, in vivo tumor growth inhibition

## Abstract

Chemotherapy is still one of the main therapeutic approaches in cancer therapy. Nevertheless, its poor selectivity causes severe toxic side effects that, together with the development of drug resistance in tumor cells, results in a limitation for its application. Tumor-targeted drug delivery is a possible choice to overcome these drawbacks. As well as monoclonal antibodies, peptides are promising targeting moieties for drug delivery. However, the development of peptide–drug conjugates (PDCs) is still a big challenge. The main reason is that the conjugates have to be stable in circulation, but the drug or its active metabolite should be released efficiently in the tumor cells. For this purpose, suitable linker systems are needed that connect the drug molecule with the homing peptide. The applied linker systems are commonly categorized as cleavable and non-cleavable linkers. Both the groups possess advantages and disadvantages that are summarized briefly in this manuscript. Moreover, in this review paper, we highlight the benefit of oxime-linked anthracycline–peptide conjugates in the development of PDCs. For instance, straightforward synthesis as well as a conjugation reaction proceed in excellent yields, and the autofluorescence of anthracyclines provides a good tool to select the appropriate homing peptides. Furthermore, we demonstrate that these conjugates can be used properly in in vivo studies. The results indicate that the oxime-linked PDCs are potential candidates for targeted tumor therapy.

## 1. Introduction

Cancer is currently the second leading cause of death globally, with 10 million deaths in 2020 worldwide. The number of new cases is expected to increase from 19.2 million to 28.4 million by 2040 [1,2]. One of the main therapeutic approaches for cancer is chemotherapy. Nevertheless, most cytotoxic agents applied in clinics for cancer chemotherapy show a narrow therapeutic window because of their inability to distinguish between the cancer and healthy cells, resulting in toxic side effects [3]. In addition, resistance to drugs may develop during the treatment, preventing complete tumor regression. To overcome these drawbacks, targeted therapeutic approaches that increase tumor selectivity have been investigated [4,5,6]. For this purpose, an anticancer drug is attached to a targeting moiety that can specifically bind to the tumor-related and/or overexpressed receptors on the cancer cells [7,8]. Two types of drug delivery systems (DDSs) are mainly applied: antibody–drug conjugates (ADCs) and small-molecule–drug conjugates (SMDCs) [9]. Among the latter ones, there is a special interest in peptide–drug conjugates (PDCs) [10]. Peptides, similarly to antibodies, can exhibit high binding affinities to the cell surface receptors; therefore, they hold a great potential as efficient homing devices for drug targeting. Although several ADCs are in clinical use to treat different types of tumors [11], PDCs provide some advantages, such as a lack of or reduced immunogenicity, better tissue penetration and an internalization capability, as well as lower costs of production, while possessing high selectivity for their target [9,10,12]. Their structures have been well characterized, and their relative drug content is higher compared to that of the ADCs. However, the PDCs are degraded and eliminated from the bloodstream quite rapidly. Despite this drawback, the mentioned benefits keep the interest in PDCs at a high level, especially the ones with increased plasma stability. In addition, the existence of many tumor types with different cell surface structures, despite deriving from the same organs, call for the treatment of a broad spectrum of cancers and the development of a large number and variety of homing devices [10].

PDCs usually consist of three parts: (i) the homing peptide that recognizes the cell surface receptor or other proteins on the cancer cells, (ii) the payload (appropriate drug molecule) and (iii) a spacer/linker that connects the homing peptide and the payload (Figure 1). The linker system plays a very important role in drug release, which may influence the anticancer activity of the conjugates. However, it is more substantial to find suitable homing peptides that can deliver the drug to cancer cells with high efficacy and selectivity. In addition, there are many different receptors on tumor cells that can be targeted by DDSs; furthermore, the receptor population varies depending on the tumors. Considering that there are many different receptors expressed on tumor cells which can be targeted by peptides, the development of numerous PDCs has significant relevance [13]. There are well-known receptors (e.g., some hormone receptors) that are overexpressed on tumor cells [14,15], but many more targets and associated homing devices have been discovered by phage display [16,17]. Nevertheless, a significant effort is required to optimize the structure of the homing peptides to increase their activity, selectivity and stability [18]. It is worth mentioning, that the synthesis of PDCs can be problematic, and because of the low yield, they might not be suitable as drug candidates [19]. To overcome this drawback, 15 years ago, we started to investigate a new structure of PDC based on an oxime linkage [20]. According to the results obtained throughout the years, we conclude that this linker strategy offers various benefits for the identification of suitable homing peptides for drug delivery. This efficient and cheap ligation method allows for the fast analysis and comparison of various homing devices. The best candidates can be selected for further optimization using other drugs and/or linkers to develop conjugates with even higher anticancer activity. The oxime-linked PDCs can also be prepared without problems in large amounts for in vivo experiments. Therefore, this ligation method provides an efficient tool for the development of PDCs for therapeutic purposes.

## 2. Linker Systems

In DDSs, the linker connects the drug with the homing motive by covalent bonding. Thus, the linkers are bifunctional compounds suitable for chemical reactions with functional groups both on the targeting moiety and on the payload (Figure 1) [4]. An appropriate linker should provide stability to the conjugate during circulation to prevent the premature release of the toxic agent, leading to off-target toxicity that might cause serious side effects. Nevertheless, the toxic payload or its active metabolite should be released in the targeted cancer cells efficiently to exert the desired antitumor effect. The PDCs enter the tumor cells mainly by receptor-mediated endocytosis, followed by accumulation in the endosomes and lysosomes, where the pH is significantly lower (4.5–6) in comparison with the extracellular environment (7.2–7.4). In addition, lysosomal enzymes like cathepsin B can take part in the degradation process. It should be considered that the linker is introduced during the synthesis process of a DDS. Therefore, the ligation reactions need to be selective, and the obtained conjugate has to be stable under further synthesis conditions and the working-up procedures. There are several applied cleavable and non-cleavable linker systems that had been well described in numerous publications. A broad overview about peptide–drug conjugates with different linkages and applications was published in 2018 by Chatzidiseri and co-workers [21]. Therefore, we discuss here only the most widely linkage systems for peptide–drug conjugates with some recently published examples (Table 1).

### 2.1. Ester Linkage

In the first peptide–drug conjugates, an ester linkage was preferred because many of drugs contain an appropriate hydroxyl group for conjugation, and it provides easy drug release. Usually, succinic or glutaric anhydrides are used to insert the linker for ester linkages. After the formation of a stable amide bond to an amino group of the peptide, the free carboxylic group is conjugated to a hydroxyl function on the drug molecule, resulting in an ester linkage. Many anticancer agents (e.g., doxorubicin (Dox), paclitaxel (PTX), camptothecin (CPT), gemcitabine and dimethylglycine etoposide) have been conjugated to homing peptides in this way [23,24,48,49,50]. The ester bond can be hydrolyzed enzymatically after internalization by the esterases in the cancer cells, and thus, release the free drug, which provides a strong antitumor effect. Furthermore, the esters can be cleaved in lysosomes by using pH-responsive linkers [22]. However, the ester bond is not fully stable in the extracellular environment; early release can occur in circulation before reaching the target cells. This might be the reason why the effect of the rather promising Zoptarelin Doxorubicin (a GnRH derivative–doxorubicin conjugate) did not significantly differ from that of free Dox in clinical trials [51]. In addition, considering that the concentration of esterases in mice blood is higher than that in human blood, esterase inhibitors have been used in preclinical studies to prevent the decomposition of the PDCs [52]. Furthermore, we observed an O-to-N acyl shift (from the aglycone part to the amino-sugar moiety in Dox) during synthesis that has been verified by the fragmentation of the glycosidic bond using tandem mass spectrometric analysis [53,54,55]. The acylation of the amino group of anthracyclines might lead to the loss of antitumor activity [56].

### 2.2. Hydrazone Linkage

Hydrazone bonds can be generated between carbonyl-containing cytotoxic agents and hydrazide-containing peptides in slightly acidic conditions. It has been written that they are stable under physiological conditions, but highly sensitive to acids and decompose under pH 5 [26]. Therefore, the hydrazone-linked drug molecule can be released easily in lysosomes, providing very good in vitro IC_50_ values, but these results might not be relevant for the prediction of in vivo data. In addition, the pH sensitivity of the hydrazone bond prevents the application of the widely applied RP-HPLC conditions, making the isolation procedure of these PDCs more difficult and providing lower yields [53]. Moreover, some studies indicated that these types of conjugates are unstable during circulation, causing substantial side effects in vivo. However, the hydrolysis rate and the corresponding stability profile of hydrazone linkers are highly variable compared to those of the other cleavable linker systems, which affect their utility [57]. Recently, doxorubicin has been the most applied drug for the development of peptide–drug conjugates with a hydrazone linkage [25,26,27,28].

### 2.3. Self-Immolative Linkers

Using self-immolative spacers is highly accepted because the free drug can be released from these constructs in tumor cells by enzymes or under reductive circumstances, while they are fairly stable in circulation. Self-immolative linkers are covalent constructs that disassemble in response to specific stimuli via intramolecular reactions that lead to stable products [58]. One prominent example is the enzyme-cleavable spacer consisting of *p*-aminobenzyl alcohol (PAB), which is often combined with enzyme cleavage site motifs, like cathepsin B-cleavable dipeptides (mainly Val-Ala, Val-Cit and Phe-Lys), that can connect different hydroxyl- or amine-containing drugs to the targeting peptides. In these structures, the amino function of PAB is acylated by the carboxyl group of the dipeptide, and the -OH group forms either a *p*-aminobenzyl carbonate or carbamate (PABC) with the drug molecule. The N-terminus of the dipeptide can be attached to amino, thiol or azido functions of the homing moiety through bifunctional agents (e.g., glutaric acid, maleimidohexanoic acid and pentynoic acid) via amide, thioether or triazole bonds, respectively [29]. After the internalization of the PDCs, the amide bond between the cathepsin B-cleavable dipeptide and PABC is enzymatically hydrolyzed in the lysosomes, followed by the 1,6-elimination of the PABC moiety, resulting in the release of the free drug. This type of spacer increases the distance between the original stimulus-responsive cleavage site and the drug, providing the efficient release of a prodrug with an instable residual spacer structure. However, the influence of a larger spacer on the receptor recognition of the homing peptide is rarely investigated.

In our recent study, GnRH derivative–drug conjugates with a self-immolative linker showed a significantly (on average, 5–10 times) lower receptor binding affinity in comparison to that of the oxime-linked conjugates, indicating that the linker structure has an impact on antitumor activity [19]. Nevertheless, this might be highly influenced by the peptide sequence and conjugation site. In addition, the development of conjugates with self-immolative spacers involves several reaction steps with low overall yields [19].

An additional widely used self-immolative linker is a disulfide-containing linker combined with ester, carbonate and carbamate for connection to the drug molecules. These linkers can be cleaved within the intracellular reducing environment by GSH, followed by thiirene release, providing the free drug [28,30,32,34].

### 2.4. Disulfide Bond

The strategy behind the use of disulfide bonds between a targeting peptide and a drug molecule relies on the reductive environment in cancer [59]. Disulfide crosslinking has been widely applied to improve the stability of drug cargos in the extracellular milieu, while promoting smooth release after the cellular uptake thanks to the 1000-fold concentration ratio of glutathione between the intracellular and extracellular environments (10 mM vs. 10 μM). Indeed, this strategy substantially stabilized the complexes in blood circulation and promoted the efficient release of drug cargos, ranging from proteins to nucleic acids [36,37]. Considering that the majority of the known anticancer drugs do not contain thiol groups, self-immolative linkers based on 1,6-elimination are combined with thiol-assisted cyclization in the applied linker system (see above) [34]. Moreover, it is worth mentioning that the stability of disulfide bonds, especially the asymmetric ones, is not always adequate [60]. Therefore, this must be monitored during in vitro biological assays and in the blood plasma. For the development of appropriate disulfide linkages, usually a bifunctional reagent (N-succinimidyl 3-(2-pyridyldithio)propionate (SPDP)) is first attached to the amino functional group on the peptide and followed by a thiol exchange reaction [38]. If possible, the application of 3- or 5-nitro-2-pyridinesulfenyl (Npys) instead of 3-(2-pyridyldithio)propionate is preferred to achieve thiol exchange in a slightly acidic condition, thus preventing the unwanted disulfide rearrangement that can happen in a slightly alkaline condition [39].

### 2.5. Amide Bond

The attachment of homing peptides through amide bond requires an amino- or carboxyl functional group on the drug molecules, such as anthracycline [19,40], methotrexate (MTX) [32,41] or chlorambucil (CLB) [42,43]. However, the biological effect of these drugs is the greatest when the functional group is free (i.e., amine on the anthracycline, carboxyl group, especially the γ-carboxyl on MTX, or on chlorambucil). The direct coupling of these two components very rarely results in conjugates with significant tumor growth inhibition because of poor drug release. The effect of the conjugates highly depends on the released metabolites, but they are rarely investigated. In the case of MTX, an additional difficulty is the selective functionalization of the appropriate carboxyl group and racemization. When MTX is randomly attached to the amino group of a peptide, mainly the γ-carboxyl group of the glutamic acid moiety acylates, affording both L-α- and D-α-isomers, but the separation of the isomers of larger peptides is problematic. It is also well known that the free amino group on the sugar moiety of anthracyclines is important to exert antitumor activity. Therefore, conjugates with adequate antitumor activity are preferably obtained by connecting these to the homing peptides via self-immolative linkers or through enzyme labile spacer (e.g., GFLG) [19,40]. Neuropeptide Y derivatives are attached to MTX directly or through either a GFLG spacer or a self-immolative spacer (disulfide/ester), releasing thiirene and free MTX after intracellular reduction [32,41]; this leads to a significantly higher bioactivity level. Böhme et al. attached CLB to an integrin-binding RGD peptide through amide-, ester- and carbamate bonds [32]. They reported that the amide bond has higher chemo- and biostability that the other two, but detected no release of the free drug, resulting in a lower in vitro antitumor activity level. Recently, daunomycin was conjugated to GnRH peptides either through an amide bond by the aid of a glutaric acid linker or by a self-immolative linker on the sugar moiety [19]. The drug release and antitumor activity of the conjugates were compared. The results demonstrated that the conjugate obtained with the amide bond could not release the free drug and had a lower in vitro antitumor activity level than the others. All these results suggest that the direct amide linkage between the homing peptides and drug molecule should not be the preferred route for the development of potential drug candidates for targeted tumor therapy.

### 2.6. Thioether Linkage

A thioether bond formed by the chemo-s elective ligation of thiol and haloacetyl groups is one of the most stable ones in both chemical and biological environments. Therefore, the direct coupling of drugs to homing peptides via such a type of thioether linkage is not often used for drug targeting. Nevertheless, the application of a thioether linkage in PDCs that can be developed easily by chemo-selective ligation, which is quite popular. However, in these cases, the applied bifunctional linkers are attached to the homing peptides through a thioether bond using Cys in the peptides, maleimide or a more reactive chloroacetyl group in the linker. The other functional group of the linker is connected to the drug with other types of bonds. Therefore, drug or metabolite release is influenced rather by this linkage, not by the thioether bond [35]. It worth mentioning that the substituted succinimide ring developed through a Michael addition reaction is not always stable and may open. In addition, the thioether bond may oxidize to form sulfoxide derivatives.

It is worth noting that in the conjugates applied for diagnosis (PET and MRI) or targeted radiotherapy, the homing peptide can be attached to the radiolabeled chelator directly through a non-cleavable covalent bond because the release from the peptide is not required for efficiency [61]. This is the case also when drug-loaded targeted nanocarriers are used for therapeutic approaches [62]. For these purposes, the application of a thioether linkage is an excellent choice because it can be developed easily by chemo-selective ligation with a good yield.

### 2.7. Oxime Linkage

The oxime bond belongs to the family of non-cleavable linkers. This type of linkage can be obtained by the reaction of a carbonyl group (either aldehyde or ketone) and an alkoxyamine functional group, but most often by an aminooxyacetylated amino group on the homing peptide. This chemo-selective click-type reaction proceeds fast in a slightly acidic buffer solution (pH 5–6) with excellent yields (almost quantitative) [20]. The oxime bond is relatively stable in a broad pH range, from pH 2 to 9 [44]. Therefore, the oximes are more stable against hydrolysis than the imines and hydrazones, but they are less stable than the triazoles formed by azide-alkyne cycloaddition and thioether linkage. The oxime linkage is also stable under physiological conditions; therefore, it has been widely used for the development of different bioconjugates, peptide–polymer conjugates, oligonucleotide conjugates, glycoconjugates, labelled bioconjugates, targeted liposomes and large proteins, by fragment assembly with chemo-selective ligation [45,46,47]. However, this is applied very rarely to PDCs because of the lack of free drug release.

In this manuscript, we provide an overview of the possibilities of applying oxime-linked PDCs for the development of drug candidates for targeted tumor therapy.

## 3. Synthesis of Oxime-Linked Peptide–Drug Conjugates

The crucial steps during the synthesis of aminooxyacetylated peptides are the incorporation of aminooxyacetic acid (Aoa), including a protecting group, and the final cleavage and the working-up procedure of the Aoa-containing peptide derivatives (Figure 1). In most cases, Boc-protected Aoa is attached to a peptide chain in the last step of solid-phase peptide synthesis [63]. It has been observed that over-acylation (additional Boc-Aoa-OH connected to the Aoa moiety) may occur as the main side reaction. Many approaches have been investigated to overcome this problem, including carbodiimide-mediated one-pot acylation without a base or the application of Boc-Aoa-OSu active ester as an acylating agent, as well as the use of a high excess (8 equiv) of Boc-Aoa-OH and coupling agents for a short acylation time (10 min) [64,65]. Nevertheless, the coupling of the diBoc-protected Aoa derivative has proved to be the best solution [66]. However, the aminooxyacetyl moiety is very sensitive to molecules containing carbonyl groups, with the partial impact of the peptide sequence. Therefore, the free NH_2_-O-R group reacts often with these compounds during the working-up procedure after the final cleavage of Aoa-modified peptides from the resin. These carbonyl group-containing derivatives might come from the plastic tubes or residues of acetone used in a laboratory. This cannot be prevented even by using diBoc-protected Aoa or working in argon. We found a highly sensitive peptide to this side reaction; the synthesis of a somatostatin analog developed in Schally’s laboratory [67] elongated with Aoa (H-Aoa-D-Phe-c[Cys-Tyr-D-Trp-Lys-Val-Cys]-Thr-NH_2_) was unsuccessful. After several trials to optimize the reaction conditions, we elaborated the following procedure: The semi-protected peptide H-D-Phe-Cys-Tyr-D-Trp-Lys(Dde)-Val-Cys-Thr-NH_2_ was cleaved from Rink Amide MBHA resin and reacted with Boc-Aoa-OPcp to incorporate Aoa into the N-terminus in a solution [68]. After the efficient coupling reaction, the Dde-protecting group was removed with 2% hydrazine in DMF. Surprisingly, during the cleavage of Dde, a cyclic peptide also formed (Boc-Aoa-D-Phe-c[Cys-Tyr-D-Trp-Lys-Val-Cys]-Thr-NH_2_). The Boc group was cleaved in 95% TFA solution in the presence of 10 equiv-free Aoa as a “carbonyl capture” reagent that could prevent the reaction of the peptide with any carbonyl derivative. The crude product was purified by RP-HPLC, and the solvent was evaporated, followed by direct ligation to daunomycin (Dau) in 0.2 M NaOAc solution at pH 5. This procedure proved to be very efficient to prepare oxime-linked Dau-peptide conjugates.

Later, isopropylidene-protected Aoa was investigated in our laboratory as an alternative to 1-ethoxyethylidene to prevent over-acylation [69]. The preparation of this compound is simple: free Aoa is dissolved in acetone, and the solvent is then evaporated. The crystalline compounds can be used directly for coupling. However, this protecting group is moderately labile to the cleavage conditions of the peptide from the resin. Depending on the sequence next to the isopropylidene-protected Aoa peptide, some unprotected compounds can be detected by HPLC. It is worth collecting this fraction as well and using it for conjugation as described above. The isopropylidene group can be removed from the lyophilized protected fraction with 1 M methoxyamine under a slightly acidic condition (0.2 M NaOAc solution at pH 5), followed by the same working-up procedure and conjugation to Dau (Figure 1).

## 4. Development of Oxime-Linked Peptide–Daunomycin Conjugate

Anthracyclines are widely used in cancer chemotherapy, and they are one of the most important derivatives in the development of PDCs. The autofluorescence properties of anthracyclines represent the main benefit for their use as payloads for our conjugates. This feature enables the investigation of cellular uptake and localization by flow cytometry and/or confocal microscopy without using an additional fluorescent dye. As a result, all the in vitro and in vivo experiments can be performed using the same batch of each conjugate. The first oxime-linked Dau and Dox conjugates were developed by Pessi et al. [70]. It was indicated that the carbonyl group in position 13 of Dau is more reactive in oxime bond formation than that of Dox. Therefore, in our research, Dau was applied for the development of oxime-linked PDCs. Chemo-selective ligation can be performed under slightly acidic conditions (e.g., 0.2 M NaOAc or NH_4_OAc buffer solution at pH 4–5) [20]. This click-type reaction provides the conjugate in a good yield (close to 100%) in few hours (depending on the peptide) in the presence of a minor excess of Dau. Therefore, the purification of the conjugate is simple because of the lack of a side product and results in a negligible loss of the final product.

### 4.1. Degradation of Oxime-Linked Peptide–Drug Conjugates in the Presence of Cathepsin B or Lysosomal Homogenate

Since the oxime bond is a non-cleavable linkage, it is important to identify the drug-containing metabolites released during the degradation of conjugates after their internalization into cells. Therefore, the proteolysis of many conjugates was investigated in the presence of cathepsin B, a lysosomal enzyme over-expressed in tumor cells and in rat liver lysosomal homogenate [71]. It was observed that the cleavage sites are similar in both the systems, but more cleavage sites were detected in the lysosomal homogenates [53,72]. Therefore, the latter one might be more relevant for these experiments. The smallest Dau-containing metabolite detected after incubation in lysosomal homogenate is Dau=Aoa-Aaa-OH, where Aaa is the amino acid of the homing peptide to which Dau is connected via the oxime linkage [72]. For instance, H-Lys(Dau=Aoa)-OH is released when the drug molecule is connected to the side chain of Lys. However, depending on the peptide sequence, sometimes this metabolite could not be observed, and the metabolites with larger peptide fragments were not active. The incorporation of a cathepsin B-cleavable spacer (e.g., GFLG or LRRY) between the homing peptide and the Aoa moiety might overcome this problem [58,73].

Because the smallest Dau-containing metabolites differ depending on the amino acid linked to Aoa; some relevant metabolites have been synthesized to investigate their DNA binding activity [72]. The structure of these metabolites and their apparent equilibrium binding constants are presented in Figure 2.

The binding affinity was lower in the case of the larger fragments. According to these results, it can be concluded that the metabolites are able to bind to DNA, but with less affinity than Dau itself. The binding efficacy depends on the amino acids remaining in the metabolite, and thus, has an impact on the antitumor activity of the conjugates. This observation suggests that different homing peptides with different amino acids on their N-terminus cannot be compared directly; therefore, the incorporation of the same enzyme-labile spacer between the homing peptide and Aoa moiety is recommended in the case of comparative studies. Conjugates providing the same metabolite allow for an appropriate comparison of the targeting efficacy of homing moieties. It is worth mentioning that the fluorescence intensity of the smallest metabolites is similar to that of free Dau [72]. Thus, the cellular uptake of free Dau and the conjugates can be followed properly.

### 4.2. Cardiotoxicity of Oxime-Linked Anthracycline–Peptide Conjugates

The main drug-related toxic side effect of anthracyclines is cardiotoxicity, leading to cardiomyopathy and heart failure [74]. Comparative experiments to determine the cardiotoxicity of peptide–anthracycline conjugates with different linkages might be informative for the other conjugates with different types of drugs as well. For this purpose, human cardiomyocytes (HCM) and human umbilical vein endothelial cells (HUVEC) were used as models. The long-term (0–72 h) cytotoxic effect of sixteen GnRH-based conjugates containing Dox and Dau was determined by real-time impedimetric sensing using the xCELLigence SP system (ACEA Biosciences, San Diego, CA, USA) [75]. The results indicated that the ester-linked GnRH-Dox conjugates, including Zoptarelin Doxorubicin, showed significant toxicity at 100 nM and 1 μM, which was remarkably pronounced on the HCM cells. The cytotoxic effect was comparable to that of the free drug, especially at the highest concentration. In contrast, the conjugates with oxime-linked Dau showed no or only a minor toxicity on both the cell lines (Table 2). These data confirm that the linkage between the payload and homing peptide has a significant influence on early drug release and, consequently, an undesired toxic side effect. We may also conclude that the search for more suitable homing peptides might be more important than the application of cleavable bonds between the drug and the peptide to develop efficient DDSs.

### 4.3. Application of Oxime-Linked Peptide–Daunomycin Conjugates for the Development of Appropriate Drug Delivery Systems

Angiopep-2 (TFFYGGSRGKRNNFKTEEY) is a peptide ligand of low-density lipoprotein receptor-related protein-1 (LRP1), which can cross the blood–brain barrier via receptor-mediated transcytosis and simultaneously target glioblastomas [76]. This peptide contains three amino groups as possible conjugation sites, and all of them have usually been used to produce PDCs [50,77]. Recently, we have studied the influence of these conjugation sites on the bioactivity of the conjugates on U87 human glioblastoma cells [78]. It was indicated that the attachment of Dau via an oxime linkage at the side chain of Lys in position 15 significantly decreases the cellular uptake rate, and hence, the cytostatic effect of the conjugate, even if the desired drug metabolite was rapidly released in lysosomal homogenate. In contrast, the modification of the side chain of Lys in position 10 provided efficient cellular uptake, but resulted in the low-level cytostatic effect of the PDC. Most probably, steric hindrance hampers the interaction with lysosomal enzymes, and subsequently, drug release. In contrast, N-terminal modification with oxime-linked Dau showed adequate cellular uptake and drug release, providing the most efficient conjugate (Figure 3). This study indicates that the antitumor activity of such conjugates depends on many factors, and the position of the drug may be more important than the number of the payloads attached. Moreover, the increase in the cellular uptake of conjugates with a substituted ^10^Lys supports the suggestion that this position might be preferred for the attachment of compounds that should not be released to cause antitumor activity (e.g., radioligands and fatty acids).

### 4.4. In Vivo Acute and Chronic Toxicity of Oxime-Linked Daunomycin–Peptide Conjugates

The most widely studied oxime-linked daunomycin–peptide conjugates were GnRH-III-based derivatives. GnRH-III (<EHWSHDWKPG-NH_2_, where <E is pyroglutamic acid) was first isolated from sea lamprey and has a very low-level hormonal effect in mammals, but binds to both the human GnRH receptors (GnRH-IR and GnRH-IIR) [79]. Since GnRH-III reveals a significant antiproliferative effect, GnRH-III derivatives may be putative homing peptides for drug delivery, especially for the treatment of hormone-independent tumors [80]. Nevertheless, the results of the in vivo acute and chronic toxicity measurements of [^8^Lys(Dau=Aoa-GFLG)]-GnRH-III were representative of the other oxime bond-linked GnRH-III-Dau conjugates with different homing peptides too. Acute toxicity was investigated either on healthy adult Balb/c female mice or on immunodeficient mice [20,81,82,83]. The experiments lasted 14 days after a single treatment. The GnRH-III conjugates were not toxic up to 30–50 mg Dau content of conjugates per kg body weight on the immunocompetent mice and 20 mg Dau content/kg on the immunodeficient mice (either i.p. or i.v. administration). In the case of the chronic toxicity studies, the appropriate dose of conjugates with 15 mg Dau content (after optimization studies) was applied using five treatments on every fourth day without any significant toxicity even on the immunodeficient mice. Nevertheless, the results of the in vivo acute and chronic toxicity measurements were representative of the other conjugates with different homing peptides too. Similarly, bombesin-based Dau conjugates were reported to be safe up to 20 mg Dau content/kg on the NSG mice in chronic toxicity studies [84]. In contrast, free Dau showed remarkable toxicity at a dose >2 mg/kg on the healthy mice and >1 mg/kg on the immunodeficient mice. The results indicated that the conjugates are not toxic even at a much higher dose than the free Dau.

### 4.5. In Vivo Tumor Growth Inhibition with Oxime-Linked Daunomycin–Peptide Conjugates

According to in vitro studies, the antitumor effect of the oxime-linked daunomycin–peptide conjugates is 1–2 orders of magnitude lower (µM range) than the efficacy of the free drug (several hundred nM), depending on the homing peptide and the tested cell line [18,83]. However, it has to be taken into account that the free drug can enter the cells by passive diffusion, while the conjugates are taken up by receptor-mediated endocytosis. Therefore, the drug concentration will always be higher in the cells in vitro in the case of the free drug. The difference might be smaller when multidrug-resistant cells are used for in vitro studies. In addition, in 3D cell cultures like spheroids, the free drug is taken up by the upper cells in the spheroids, while the conjugates can reach the cells that are closer to the core [85]. Nevertheless, while in vitro studies are suitable for the comparison of different conjugates, the real applicability of these conjugates can only be answered by in vivo experiments.

For in vivo experiments, the tumors were either inoculated subcutaneously (s.c., the most widely used protocol) or orthotopically (breast and colon tumors) in mice [20,81,82,83,86]. In some cases, the conjugates that showed significant tumor growth inhibition on the s.c.-developed tumors were not potent on the orthotopically developed ones [82]. Therefore, orthotopic implantation is preferred if it is possible. In most cases, the conjugates showed higher tumor growth inhibition at the applied dose than the free drug at the maximum tolerated dose. Moreover, the conjugates displayed significantly less toxicity and resulted in the longer survival of the animals in comparison with free Dau administration. The mice treated with Dau had to be sacrificed earlier in many cases because of their significant weight loss. The conjugates also showed higher-level antimetastatic effects and a lower proliferation index (reflects the extent of the proliferative activity of tumor cells) and the vascularization of the tumor tissues in comparison with those of free Dau. Some examples are summarized in the following paragraphs.

### 4.6. In Vivo Antitumor Effect of Oxime-Linked GnRH-III Derivative–Dau Conjugates

In our experiments, GnRH-III was applied as a targeting peptide. The conjugate [^8^Lys(Dau=Aoa-GFLG)]-GnRH-III, in which Dau was connected to the side chain of Lys in position 8 through an aminooxyacetylated cathepsin B-labile GFLG spacer, showed significant tumor growth inhibition in s.c.-developed fast-growing C26 murine colon cancer-bearing Balb/c mice [20]. The effect highly depended on the treatment schedule. In the first attempt, the conjugate was administered i.p. at a dose of 8.86 μmol (5 mg Dau-content/kg body weight) five times on days 7, 9, 11, 14 and 16 after tumor transplantation (Treatment schedule A). The tumor volume inhibition was only 15.7% on day 26 when the experiment finished. A slight enhancement in inhibition (22.3%) was detected when a dose of conjugate corresponding to 15 mg Dau content/kg body weight (26.6 µmol) was injected only once on day 7 (Treatment schedule B). An additional treatment on day 10 (Treatment schedule C) did not result in any further improvements (21.9% on day 29). However, when the treatment schedule was changed to two treatments with the same dose on days 4 and 7, 46.3% inhibition was observed on day 29 (Treatment schedule D). In contrast, the treatments with free Dau at a dose of 2 mg/kg body weight (3.55 μmol) on days 7, 9, 11, 14 and 16 showed only 22.6% inhibition on day 26. The median survival rates of the treated animals in comparison to the control group were 1 (A), 1.23 (B), 1.21 (C) and 1.38 (D), respectively, and 0.81 for free Dau. These results indicate the lower toxicity and the higher tumor volume inhibition effect of the conjugates in comparison with those of free Dau, as well as the importance of the treatment schedule.

In another experiment, HT-29 human colon cancer was developed s.c. in immunodeficient SCID mice. Dau and two conjugates (with or without a GFLG spacer between the GnRH-III homing peptide and the payload) were used for the treatment [81]. The first i.p. administrations were performed on day 13 after tumor inoculation. All the mice treated once with 2.5 mg/kg body weight (4.43 μmol) free Dau died within 10 days. In contrast, the conjugates at a dose of 15 mg Dau content/kg body weight (26.6 μmol) that refers to 52 mg/kg [^8^Lys(Dau=Aoa)]-GnRH-III and 62.5 mg/kg [^8^Lys(Dau=Aoa-GFLG)]-GnRH-III conjugates, respectively, did not show significant toxicity. The treatment was repeated on days 23 and 30. Because of the significant weight loss in several mice in the control group, the experiment was terminated on day 35. The tumor growth inhibition could be calculated as reductions in the tumor volume by 44.3% and 57.6% and the tumor weight by 41% and 50%, respectively.

Interestingly, the conjugates were poorly effective on orthotopically developed tumors [82,86]. In the case of the C26 colon tumor-bearing female Balb/c mice, only a 7% reduction in tumor weight was detected on day 13 after the two treatments on days 4 and 7 with [^8^Lys(Dau=Aoa)]-GnRH-III, at a 26.6 μmol/kg (15 mg Dau content) dose. The effect of free Dau (2 mg/kg on days 4 and 7) showed a better inhibitory effect (24.4%). Interestingly, our novel developed GnRH-III derivative, in which Ser in position 4 was replaced by Lys(Ac), was much more potent, with 49.3% inhibition. It is worth mentioning that the rate of cellular uptake of the [^4^Lys(Ac), ^8^Lys(Dau=Aoa)]-GnRH-III conjugate by the tumor cells was significantly higher than that of the conjugate with the native GnRH-III sequence (Table 3).

Starting from these promising results, Dau conjugates with further GnRH-III derivatives using different short-chain fatty acids (on the side chains of Lys in position 4) were developed [73]. The comparative in vitro experiments suggested that the most efficient one based on the cytostatic effect, cellular uptake and stability is the butyric acid containing conjugate ([^4^Lys(Bu), ^8^Lys(Dau=Aoa)]-GnRH-III. Therefore, this compound was tested in vivo and compared to the acetylated version and free Dau [82]. Their tumor growth inhibition was investigated on orthotopically developed HT-29 human colon cancer-bearing immunodeficient mice. The treatment schedule was as follows: The first i.v. treatment was given on day 5 after inoculation with a conjugate dose of 15 mg Dau content (26.6 μmol)/kg, followed by four additional i.v. treatments twice a week. Afterwards, eight treatments at a maintenance dose of 7.5 mg Dau content (13.3 μmol)/kg were administered i.p. twice a week. Free Dau was injected i.v. once a week using 1 mg/kg dose for 7 weeks. The [^4^Lys(Ac), ^8^Lys(Dau=Aoa)]-GnRH-III conjugate did not show any significant tumor growth inhibition effect (7.1% reduction in tumor weight) on this model, while [^4^Lys(Bu), ^8^Lys(Dau=Aoa)]-GnRH-III had a significant effect, with 39.4% inhibition, which is higher than the activity level of free Dau (29.7%). In addition, free Dau showed liver toxicity (11% relative liver weight loss in comparison with the controls), while the conjugates significantly decreased the proliferation index and vascularization rate compared to free Dau and to the control. The proliferation indexes on average were 86.3% for [^4^Lys(Ac), ^8^Lys(Dau=Aoa)]-GnRH-III, 84.5% for [^4^Lys(Bu), ^8^Lys(Dau=Aoa)]-GnRH-III and 98.6% for Dau, respectively, compared to the controls, while the detected vascularization rates were 64.8%, 67% and 112.5%, respectively. The conjugates more significantly decreased metastasis in pancreas and lymph nodes compared to that of the control, but not to Dau. No positive effect was detected in the lungs, livers or spleens.

The further optimization of the GnRH-III sequence lead to the discovery of ([^2^ΔHis, ^3^D-Tic, ^4^Lys(Bu), ^8^Lys(Dau=Aoa)]-GnRH-III as the most promising conjugate [87]. Here, the His-Trp dipeptide fragment was replaced by D-Tic (D-tetrahydroisoquinoline-3-carboxylic acid), a non-native amino acid. The superior in vitro cytostatic data (below 1 μM) could be explained by the enhanced cellular uptake of this conjugate (Figure 4).

The antitumor effects of the previous ([^4^Lys(Bu)], ^8^Lys(Dau=Aoa)-GnRH-III) and new lead compounds ([^2^ΔHis, ^3^D-Tic, ^4^Lys(Bu), ^8^Lys(Dau=Aoa)]-GnRH-III) were compared on SCID mice with orthotopically inoculated HT-29 colon cancer [82].

The animals were treated seven times (twice a week) with 17.72 μmol conjugates (10 mg Dau content)/kg body weight s.c. on days 7, 10, 13, 16, 20, 23 and 27. Free Dau was administered at a 1 mg/kg dose on days 7, 13 and 20. The experiment finished on day 30, but the animals in the group treated with Dau had to be sacrificed on day 23. The tumor weight was reduced by 80.8% and 87.1% by the conjugates, respectively, while the value was 84.3% for Dau. However, these numbers cannot be directly compared because the termination days were different during the experiment. The toxicities of the compounds were evaluated as the rate of changes in the liver and body weights (99.2% and 89.0% for the previous and the new conjugates, and 70.6% for Dau, respectively). All these data also confirm that the conjugates can be used at significantly higher doses, providing a similar or even enhanced antitumor activity and fewer toxic side effects compared to the free drug.

## 5. Development of Bombesin-Based Peptide–Drug Conjugates

Like GnRH and somatostatin, bombesin (BBN) is another example of peptide hor-556 mone which receptors are overexpressed in cancerous tissues. This 14-mer peptide (Glp-Gln-Arg-Leu-Gly-Asn-Gln-Trp-Ala-Val-Gly-His-Leu-Met-NH_2_), first discovered in the skin of the frog *Bombina bombina*, is associated with the gastrin-releasing peptide (GRP) as a mammalian counterpart [88,89]. The bombesin receptor family comprises neuromedin receptor B (NMB-R, or BB1), gastrin-releasing peptide receptor (GRP-R, or BB2) and bombesin receptor subtype 3 (BRS-3, or BB3). Among them, GRP-R has been the most investigated so far and has been proven to be upregulated in breast, prostate, pancreas, small-cell lung cancers, among others, hence representing a suitable target for drug delivery to tumors [90,91,92]. Many attempts have been made to modify and shorten the sequence of bombesin to tailor its stability, activity (agonist or antagonist), affinity and selectivity towards this receptor [92,93,94,95,96,97,98]. Several research groups have encouraged the use of their optimized structures as putative drug delivery systems, but they were never directly compared. Moreover, the previous examples of conjugates between GRP-R ligands and anthracyclines display a labile ester bond that could cause the early release of the drug in vivo [91,99]. Therefore, our group took the leap and produced conjugates based on the most promising bombesin analogs as homing devices attached to Dau via an oxime bond and two cathepsin B-cleavable linkers (LRRY or GFLG) [84]. Furthermore, a conjugate bearing a novel developed bombesin analog ([^6^D-Phe, ^11^β-Ala, ^13^Sta, ^14^Nle]-BBN(7–14)) was synthesized. The use of oxime-linked Dau as a payload promoted the identification of three conjugates with improved cytostatic activity and cellular uptake by human prostate (PC-3) and breast cancer (MDA-MB-231 and MDA-MB-453) cell lines. The Dau=Aoa-Leu-OH active metabolite was readily released in all the cases in less than 30 min in rat liver lysosomal homogenate, but only L5 (Dau=Aoa-LRRY-[^6^D-Phe, ^13^Sta, ^14^Leu]-BBN(7–14), where Sta is statine) and L6 Dau=Aoa-LRRY-[^6^D-Phe, ^11^β-Ala, ^13^Sta, ^14^Nle]-BBN(7–14) demonstrated a satisfactory stability in the mouse plasma (Table 4). Therefore, only these two conjugates were further investigated in vivo. As mentioned above, their chronic toxicity was assessed in healthy mice by administering the conjugates doses of 5, 10 and 20 mg Dau content/kg body weight. The PDCs were not critically harmful at any dose, although a relatively high mouse weight loss (10–15%) was induced at the highest dose. The in vivo antitumor efficacy was studied in murine xenograft models bearing s.c.-inoculated PC-3 human prostatic adenocarcinoma. The conjugates were administered intraperitoneally every fifth day starting from day 9 after tumor inoculation, for a total of five treatments, at a dose of 10 mg Dau-content/kg body weight, and their tumor growth inhibition values were compared to those of the mice treated with the maximum tolerated dose of free Dau (1 mg/kg) and 0.9% saline solution (control group). On the last day of the experiment (day 33), L5 and L6 revealed reductions in the tumor size by 21.4% and 31.4% and tumor weights by 16.6% and 33.1% compared to those of the control, respectively (Table 4). On the other hand, the Dau-treated group had to be terminated on day 26 because of severe toxicity, without showing a significant reduction in the tumor volume and weight. To better understand the long-term effect of the newly developed compounds, the tumor doubling time (DT) was calculated; both the PDCs significantly increased the DT compared to the treatment with free Dau.

As a result, only two among the many bombesin analogs could be identified as the most suitable for tumor targeting through direct in vitro comparison by using the oxime linkage strategy, but it was also demonstrated that the Dau conjugates carrying them as homing device could improve the selectivity and reduce the toxicity of the free drug in vivo.

## 6. Improved Antitumor Activity of Selected Homing Peptide from Phage Display Library by Sequence Optimization

Many potential homing peptides for selective drug targeting are selected from phage display libraries. However, their selectivity and receptor-recognizing capability highly depend on the applied phage display technique (in vitro, in vivo or ex vivo) and the number of selection cycles [16,100,101]. A larger number of selection cycles can lead to more potent homing peptides. Nevertheless, we believe that the optimization of the selected homing peptides might lead more potent carriers for targeted therapy.

### 6.1. Phage Display-Based Homing Peptide Linked to Daunomycin via Oxime Bond for Selective Drug Targeting of HT-29 Colon Cancer

In our group, we synthesized conjugates based on a heptapeptide (VHLGYAT) selected by Zhang et al. and tested them on HT-29 human colon adenocarcinoma [102]. Since sequence modification may result in different oxime-linked Dau-amino acid fragments, a cathepsin B- labile spacer was incorporated to the N-terminal of the homing peptide [18]. From the parent compound (Dau=Aoa-LRRY-VHLGYAT-NH_2_), the potential positions that can be substituted were selected using Ala-scan. The replacement of Gly with Ala resulted in a conjugate with higher-level cytostatic effect, thanks to the enhanced cellular uptake of the modified compound. In the second optimization step, different amino acids were incorporated in this position. The in vitro experiments indicated that apolar amino acids with bulky side chains (Leu, Phe and Phe(pCl) provide conjugates with significantly higher activity levels. For further studies, Dau=Aoa-LRRY-VHLFYAT-NH_2_ was selected and compared with the parent conjugate. The in vitro cytostasis studies showed a 2–5 times higher activity level on the different cancer cell lines, e.g., up to 2.6 times higher on HT-29. Orthotopically developed HT-29 tumor-bearing SCID mice were used for the in vivo experiments. One group of mice was treated with free Dau (1 mg/kg body weight) on days 13 and 20 after tumor transplantation. The groups that received the conjugates were treated with a dose of 10 mg/kg Dau content (36.6 and 38.3 mg/kg of each conjugate, respectively) on days 13, 16, 20, 23 and 27 after tumor transplantation using i.p. administration. The experiments were terminated on day 30, but the animals treated with free Dau were sacrificed on day 23 because of significant weight loss. The parent compound and the Phe-containing conjugate reduced the tumor weight by 65% and 89% compared to that of the control group, respectively (Figure 5a). The free drug inhibited tumor growth by 84% compared to that of the control, but it has to be pointed out that their end points were shifted by one week. The toxicity of the free drug was also indicated by the significant decrease in the liver weight (28% lower than the controls), while it was inferior to 10% in case of the conjugates (Figure 5b). The proliferation index was significantly decreased (42% for the control) by the conjugate with an optimized structure (Figure 5c).

By the aid of this new homing peptide, characterized by a higher affinity than the parent compound, we could identify the most probable target receptor using affinity chromatography, HPLC-MS and a protein database: heat shock protein 70 (Hsp70). The Hsp70-positive tumor phenotype is associated with the aggressiveness and therapy resistance of cancer, and membrane-bound Hsp70 plays a pivotal role in eliciting an antitumor immune response [18,103,104].

### 6.2. Phage Display-Based Homing Peptide Linked to Daunomycin via Oxime Bond for Selective Drug Targeting of PANC-1 Pancreatic Cancer

Several homing peptides that recognize pancreatic cancer cells and are used for drug targeting directly or as a part of nanoparticles have been described in the literature [105,106,107,108]. One of them is the CKAAKNK oligopeptide, which has been selected by phage display and can specifically bind to the tumor vessels in RIP-Tag2 transgenic mice, a prototypical mouse model of multistage pancreatic islet cell carcinoma [109].

We assessed a modified version of this peptide as a homing device, where Cys was replaced by Ser because the thiol group is not necessary to introduce an oxime linkage. Among the prepared conjugates, Dau=Aoa-GFLG-K(Dau=Aoa)SKAAKN-OH (Figure 6) showed the highest cellular uptake and cytotoxic activity levels in vitro on the PANC-1 cells [110]. Using this conjugate at 10 μM concentration, the cell viability of the PANC-1 cells was 0.1% after a 72 h treatment. In contrast, the cell viability of normal human dermal fibroblast (NHDF), a model of healthy cells, was roughly 75%, showing a significant selectivity for tumor cells (Figure 7). From this conjugate, three drug containing metabolites were released (Dau=Aoa-Gly-OH, Dau=Aoa-Gly-Phe-OH and H-Lys(Dau=Aoa)-OH) in the lysosomal homogenate within 72 h (Figure 6).

The in vivo tumor growth inhibitory effect of the conjugate and free Dau was investigated on s.c.-developed PANC-1 tumor-bearing SCID female mice. The free drug was applied once a week at a maximum tolerated dose of 1 mg/kg body weight (three treatments), while the conjugate was administered, on average, three times per a week either at a 2 mg/kg or 10 mg/kg dose, calculated as per the Dau content (eighteen treatments altogether). The first treatment was given on day 10 after tumor inoculation. After the third treatment with free Dau, the animals lost 20% of their weight; therefore, the mice in this group were sacrificed on day 28. The toxic effect of the free drug was detected also by the significant decrease in liver weight (29.1%). During this time, no significant inhibition of tumor growth was detected in any group. The experiment with the control group and the groups treated with the conjugate was terminated on day 74. During this time, the tumor growth inhibition by tumor volume was 14% in the case of 2 mg/kg (Dau content) dose and 32.2% when the 10 mg/kg (Dau content) dose was applied. It is worth mentioning that the highest inhibition rate was observed on day 67 (after 16 treatments), when the tumor was reduced by 43% compared to that of the control. This might be explained by the development of tumor resistance after a while. No toxic side effects could be observed in the mice treated with the conjugate. The tumor growth inhibition was also evaluated by measuring the difference in tumor weight, which showed reductions of 27.3% and 30.4% when using 2 mg/kg and 10 mg/kg doses, respectively. In summary, the data indicated that the conjugate developed for pancreatic cancer targeting inhibited the tumor growth of s.c. human pancreatic cancer (PANC-1)-bearing mice significantly without causing any toxic side effects.

## 7. Other Examples of Oxime-Based Conjugates for Targeted Tumor Therapy

The beneficial properties of the oxime linkage have recently been leveraged for tumor therapy in a variety of other conjugates, which display other peptides [4,111,112], antibodies [113,114,115,116,117,118,119,120], polysaccharides [121] or 19-nortestosterone [122] as carriers. We would like to draw attention to two particular products that have reached clinical trials: ASP1235 (previously AGS62P1; Phase I—NCT02864290) by Astellas Pharma [14,115] and ARX788 by Ambrx (Phase II trials ongoing) [116,117,118,119,120]. Both are ADCs consisting of a proprietary microtubule-targeting agent as a cytotoxic payload attached to an IgG-like antibody. The former targets FLT3 (FMS like tyrosine kinase-3) to treat acute myelogenous leukemia (AML), while the latter targets HER-2 (human epidermal growth factor receptor 2)-expressing cancers, such as breast and gastric neoplasms. Both these monoclonal antibodies are engineered to express a residue of p-acetyl-phenylalanine (pAF) in two specific positions on the heavy chains to allow for the conjugation of the toxin via an oxime bond with defined stoichiometries (drug–antibody ratio (DAR) = 2). This oxime-based site-specific conjugation technology offers a platform to produce highly stable products in circulation [114,116], improving the efficacy and controlling the off-target toxicities frequently observed in the first generation of heterogeneous ADCs [123]. The cytotoxic drug is only released as an active metabolite consisting of pAF attached to the payload through the intact oxime bond after internalization in the cancerous cells and the consequent endosomal and lysosomal proteolysis of the ADC [114,116]. This metabolite has been reported to be well tolerated in rats and cynomolgus monkeys [114]. As a result, ASP1235 and ARX788 demonstrated promising preclinical safety and efficacy profiles that led to further clinical development. ASP1235 was evaluated as a monotherapy in a Phase I clinical trial (NCT02864290) and as a combination together with venetoclax and azacitidine to treat AML. It displayed significantly greater tumor regression compared to both the ADC alone and the combination of venetoclax and azacitidine, as well as a safer profile (doses 3 mg/kg, 3 mg/kg and 100 mg/kg, respectively) [115]. ARX788 demonstrated superior efficacy compared to that of the FDA-and EMA-approved trastuzumab emtansine (T-DM1) in mice bearing HER-2 positive HCC1954 breast and NCI-N87 gastric cancer xenografts [116] and in HER-2-positive or low-expression-level HER-2 breast and gastric tumor cell- and patient-derived xenograft models [117,118]. The improved tumor growth inhibition and safety profile can be ascribed to the in vivo stability of the oxime bond that limits the premature and non-specific release of payload, enriching its concentrations at the tumor site. In a Phase I clinical study, ARX788 was administered to 69 metastatic breast cancer patients receiving anti-HER-2 therapies at doses of 0.33, 0.66, 0.88, 1.1, 1.3 or 1.5 mg/kg every 3 weeks, or 0.88, 1.1 or 1.3 mg/kg every 4 weeks [119]. No dose-limiting toxicities or drug-related deaths occurred, while the administration of 1.5 mg/kg ADC every 3 weeks resulted in an objective response rate of 65.5%, a disease control rate of 100% and a median progression free survival (PFS) of 17 months. Moreover, grade ≥ 3 adverse events (AEs) occurred with a frequency as low as 11.6%, of which only 1.4% were hematological toxicities that are commonly observed in other ADCs [123]. Another Phase I study involved 30 participants with either gastric adenocarcinoma or gastroesophageal junction adenocarcinoma [120]. They were divided into three groups, receiving ARX788 at doses of either 1.3 mg/kg, 1.5 mg/kg or 1.7 mg/kg every 3 weeks to evaluate the safety and tolerability as a primary endpoint and the preliminary efficacy as a secondary endpoint. The conjugate was well tolerated, with only four patients incurring grade 3 AEs, no grade 4 or 5 events occurred. To note, grade ≥ 3 neutropenia was never observed. Furthermore, the antitumor activity of ARX788 was promising, with an ORR of 37.9%, an overall survival of 4.1 months and a median PFS of 10.7 months. These encouraging preclinical and clinical data regarding the safety and efficacy of ARX788 are consistent with the advantageous presence of the oxime linkage in tumor-targeting conjugates. Its enhanced stability is the basis of the reduced off-target toxicities and the improved distribution of the cytotoxic payload at the tumor site. This is also likely the reason why the great incidence of grade ≥ 3 neutropenia observed with many ADCs, such as trastuzumab deruxtecan, is significantly reduced with ARX788.

All these data also indicated that the oxime-linked PDCs or ADCs have potential perspectives as potential chemotherapeutics in personalized targeted tumor therapy.

## 8. Conclusions

The development of peptide–drug conjugates for targeted tumor therapy is an important research topic. Among the parts of the conjugates, the role of the homing peptide is of great significance in providing the desired activity. Personalized targeted tumor therapy might require a huge number of homing devices because the receptor profiles of tumor cells might be significantly different. In addition, the number of receptors on tumor cells is limited; therefore, a combination of conjugates might be necessary to reach an appropriate concentration of the drug in the tumor cells. In our research, we described a simple method to develop peptide–drug conjugates that can be compared easily. The production of these conjugates is straightforward, with high yields, affording the appropriate amounts of compounds also for in vivo assays. The autofluorescence of anthracyclines offers a useful tool to detect the cellular uptake of the conjugates by tumor cells and the localization of drug metabolites in the cells. The same conjugates can be applied for in vitro and in vivo experiments, like cytostatic and cytotoxic effects, cellular uptake, acute and chronic toxicity studies and tumor growth inhibition. In this review, we have also reported that peptide–anthracycline conjugates are not only ideal for the selection of suitable homing peptides, but can also provide a significant in vivo antitumor effect that demonstrates their potential as future drugs. Furthermore, the hereby selected homing peptides can be used for the development of drug candidates in combination with other types of payloads.

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
