# Peer review of "Oxime-Linked Peptide–Daunomycin Conjugates as Good Tools for Selection of Suitable Homing Devices in Targeted Tumor Therapy: An Overview"

_ijms, 2024, doi:10.3390/ijms25031864_

Round 1
Reviewer 1 Report
Comments and Suggestions for Authors
Gábor Mező and team presented a review topic entitled “Oxime-linked peptide–daunomycin conjugates as good tools for selection of suitable homing devices in targeted tumor therapy: An overview”. The team mainly focused on advancements of oxime-linked anthracycline peptide conjugates in developing peptide-drug conjugates. The authors emphasized the benefits and disadvantages of the oxime-linked peptide–drug conjugates along with other types of crosslinking agents. Overall, the manuscript is well-framed and can be accepted for publication in MDPI’s Journal: International Journal of Molecular Sciences.
1. Table 1 and Section 2.3: The introduction of para-aminobenzyl carbonate or carbamate groups following disulfide also makes self-immolation after reducing the disulfide (-SS-) bonds to thiol (SH) groups. Hence, we recommend the authors change the self-immolation in Table 1 and Section 2.3.
2. The suggested references will help to emphasize the rationale for incorporating disulfide linkers between drugs and scaffolds. For example, disulfide crosslinking has been widely applied to improve the stability of drug cargoes in the extracellular milieu while promoting smooth release after intracellular entry because reversibly crosslinks between the drug and scaffold are cleavable in the intracellular space (10 µM), where the glutathione concentration is 1000-fold higher than that in extracellular space (10 mM). Indeed, this strategy substantially stabilized the complexes in the blood circulation and promoted the efficient release of drug cargoes ranging from proteins and nucleic acids (ACS Nano 2020;14(6):6729-6742and J Drug Target 2019;27(5-6):670-680). Please include the rationales for every crosslinker like this.
3. Define the acronyms or abbreviations after their first use in the manuscript. For example, Page 5, Line 201: SPPS?
4. Remove unnecessary abbreviations or acronyms if the scientific information is given only once. For example, on Page 7, Line 246: FACS
5. It may be interesting to expand the review to include peptide drugs and other therapeutics, such as messenger RNA (a hot topic of vaccines) and other nucleic acids.
6. Page 2, Line 49: Discuss the rationale behind the particular interest in peptide drug conjugates immediately after the sentence.

Author Response
The responses to the Reviewer was attached.

Reviewer 2 Report
Comments and Suggestions for Authors
The manuscript is a overview of a large number of interesting and in-depth works done by the authors related to the development of strategies for obtaining of anthracycline conjugates as an example of daunomycin with targeting peptides using the oxime-based pathway, as well as verify of the strategy in experiments in vitro and in vivo. The manuscript may be published in International Journal of Molecular Sciences, but after solving a number of issues.
1. It might be interesting to add a section with examples of targeting peptides currently used to develop materials for diagnostics and therapeutics.
2. In Table. 1 for better visualization, it can be added imaging of the formed bonds under the name of linkage in the “Linkage” column (for example, Ester -C(=O)O-, etc.). In the “Used functional groups” column, provide only general formulas, avoiding specific reagents. So, for example, in the line “Thioether”, instead of MCA, it can be used the formula of only the maleimide group, since other maleimide-containing derivatives can be used instead of MCA. Also it can be added a “References” column.
3. In Section 2.5 would add more references that describe the preparation of targeting peptide conjugates using maleimide-containing linkers (for example see https://doi.org/10.1021/acsomega.1c05815 and https://doi.org/10.1016/j.colsurfb.2022.112981).
4. Can the authors give examples of drug-linker-peptide conjugates linked via an amide bond?
5. The manuscript contains subsection “6.1. A house-made phage display-based peptide linked to daunomycin via an oxime linkage for selective targeting of PANC-1 pancreatic cancer.” It probably needs to be separated into a section “7”.
Author Response
The responses to the Reviewer is attached.

Reviewer 3 Report
Comments and Suggestions for Authors
The manuscript by Mező and co-authors presents a literature review on using peptide-drug conjugates as drug delivery systems with anti-tumor properties. Different types of linkers connecting small drug molecules and homing peptides are analyzed with a special emphasis on the oxime-based linkers. It was shown that the use of oxime linkers is promising for the development of systems for targeted drug delivery in cancer chemotherapy. The literature cited is relevant and up-to-date. The manuscript deserves publication.
Specific comments:
Table 1: The footnotes (explanations for PABC, MCA, etc.) should be properly referenced from the table using asterisks or superscript letters. The same refers to tables 2-4.
Lines 101-102: Possibly, the compound name should be corrected as "6-maleimidocaproic acid". Please check also throughout the manuscript.
Line 201: The abbreviation "SPPS" should be explained on its first mentioning.
Figure 2: The resolution of the figure is very low and should be enhanced.
Please use italic font for "in vitro" and "in vivo" throughout the text.
Summarizing, I recommend acceptance of the manuscript for publications after minor revision.
Author Response

(The authors gave the same response as above.)

Round 2
Reviewer 1 Report
Comments and Suggestions for Authors
The authors adequately revised the manuscript. Hence, the current version can be considered for publication.